# SEEING THE UNSEEN: HOW EMoE UNVEILS BIAS IN TEXT-TO-IMAGE DIFFUSION MODELS

## ABSTRACT

Estimating uncertainty in text-to-image diffusion models is challenging due to their massive parameter counts (often exceeding 100M) and operation in complex, high-dimensional spaces with virtually unbounded input domains. We introduce Epistemic Mixture of Experts (EMoE), a framework for efficient estimation of epistemic uncertainty in diffusion models. EMoE leverages pre-trained networks without requiring additional training, enabling direct uncertainty estimation from a prompt. By probing a latent space within the diffusion process, EMoE captures epistemic uncertainty more effectively than existing approaches. Experiments on the COCO dataset demonstrate EMoE's superior performance. Beyond benchmark gains, EMoE highlights under-sampled languages and geographic regions associated with elevated uncertainty, uncovering hidden biases in training data. Since training data for online diffusion models is rarely made public, this bias-detection capability is especially valuable. Together, these contributions position EMoE as a practical tool for addressing data imbalance and improving inclusivity in AI-generated content.

## 1 INTRODUCTION

In recent years, text-to-image diffusion models have achieved remarkable progress, enabling faster generation (Song et al., 2020; Liu et al., 2023; Yin et al., 2024), higher visual fidelity (Dhariwal and Nichol, 2021; Nichol et al., 2022; Rombach et al., 2022), and even video synthesis (Ho et al., 2022b; Khachatryan et al., 2023; Bar-Tal et al., 2024). These models operate via a forward process that gradually adds noise to data and a reverse process that learns to denoise and reconstruct it. Despite their success, diffusion models remain largely black boxes: they provide little transparency about their uncertainty or the data they were trained on (Berry et al., 2024; Chan et al., 2024).

We address this limitation with Epistemic Mixture of Experts (EMoE), a framework for capturing and quantifying epistemic uncertainty in text-conditioned mixture-of-experts diffusion models. Epistemic uncertainty, arising from a model's lack of knowledge, is reducible with additional data—unlike aleatoric uncertainty, which reflects irreducible noise (Hora, 1996; Der Kiureghian and Ditlevsen, 2009; Hüllermeier and Waegeman, 2021). As such, epistemic uncertainty offers a powerful means of detecting biases and underrepresented regions in training data that remains hidden from public access.

Figure 1 illustrates this phenomenon. For the English prompt "Two teddy bears are sitting together in the grass", EMoE reports low epistemic uncertainty (0.38). The same prompt in Finnish ("Kaksi nallekarhua istuu yhdessä nurmikolla") produces much higher uncertainty (0.83). This disparity highlights how models trained predominantly on English exhibit degraded performance for low-resource languages, reinforcing inequities in multimodal AI.

The EMoE framework rests on two core components. First, it leverages pre-trained mixture-of-experts (MoE) models for zero-shot uncertainty estimation. The experts in the MoE were independently trained on different data subsets, and EMoE harnesses their diversity without additional training. This design yields ensemble-like benefits at a fraction of the computational cost of training diffusion ensembles from scratch, which otherwise require hundreds of GPU-days (Balaji et al., 2022). Second, EMoE estimates uncertainty in the latent space of the denoiser, enabling early detection of under-sampled prompts before image generation is complete. This latent-space analysis allows users to halt expensive denoising when uncertainty is prohibitively high (Song et al., 2024).

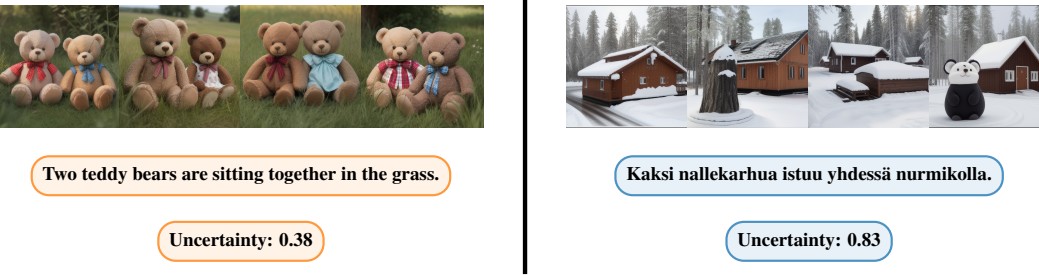

Figure 1: This figure presents the model's uncertainty estimates when interpreting the same prompt in two languages. On the left, the English sentence "Two teddy bears are sitting together in the grass." yields a lower uncertainty score of 0.38, indicating greater model confidence. On the right, the corresponding Finnish translation "Kaksi nallekarhua istuu yhdessä nurmikolla." results in a higher uncertainty of 0.83, reflecting reduced confidence. This contrast highlights how the model responds differently to in-distribution (English) versus out-of-distribution (Finnish) language inputs.

We evaluate EMoE on the COCO dataset (Lin et al., 2014) and summarize our contributions as follows:

- We introduce EMoE, a framework that combines pre-trained mixture-of-experts with latent-space variance to estimate epistemic uncertainty in diffusion models.
- We demonstrate that EMoE produces uncertainty estimates consistent with expectations, validated on high-performing MoE diffusion models using the COCO dataset.
- We show that EMoE detects novel or underrepresented data by quantifying uncertainty across 25 languages, revealing systematic biases even when the training data is not publicly available.
- We provide ablation studies that analyze design choices and confirm the robustness of EMoE's components.

Our results shed new light on epistemic uncertainty in text-conditioned diffusion models and establish EMoE as a practical tool for building more transparent and inclusive generative AI systems.

## 2 BACKGROUND

Diffusion models iteratively add and remove Gaussian noise, forming a Markov chain to generate samples. This structure naturally supports uncertainty estimation, as probability distributions inherently model uncertainty (Hüllermeier and Waegeman, 2021). Furthermore, since ensembles are commonly used to estimate epistemic uncertainty (Hoffmann and Elster, 2021), MoE models are well-suited for this task by creating an ensemble of experts.

### 2.1 DIFFUSION MODELS

In the context of supervised learning, consider a tuple $(x, y)$, where $x \in \mathbb{R}^{512 \times 512 \times 3}$ and $y$ is the prompt associated with the image. The objective is to estimate the conditional distribution $p(x|y)$. However, due to its high-dimensional and multi-modal nature, some diffusion models operate in a latent space learned by an autoencoder (Rombach et al., 2022). The autoencoder consists of an encoder $\mathcal{E}$, which maps images to their latent representation, and a decoder $\mathcal{D}$, which reconstructs images.

Diffusion models use a two-phase approach, consisting of a forward and a reverse process, to generate realistic images. In the forward phase, an initial image $x$ is encoded to $z_0$ and then gradually corrupted by adding Gaussian noise over $T$ steps, resulting in a sequence of noisy latent states $z_1, z_2, \ldots, z_T$. This process can be expressed as:

$$q(z_t|z_{t-1}) = \mathcal{N}(z_t; \sqrt{1 - \beta_t} z_{t-1}, \beta_t \mathbf{I}) \qquad q(z_{1:T}|z_0) = \prod_{t=1}^{T} q(z_t|z_{t-1}),$$

where $\beta_t \in (0, 1)$, with $\beta_1 < \beta_2 < \cdots < \beta_T$. This forward process draws inspiration from non-equilibrium statistical physics (Sohl-Dickstein et al., 2015).

The reverse process aims to remove the noise and recover the original image, conditioned on text. This is achieved by estimating the conditional distribution $q(z_{t-1}|z_t, y)$ through a model $p_\theta$. The reverse process is defined as:

$$p_\theta(z_{0:T}|y) = p(z_T) \prod_{t=1}^{T} p_\theta(z_{t-1}|z_t, y) \qquad p_\theta(z_{t-1}|z_t, y) = \mathcal{N}(z_{t-1}; \mu_\theta(z_t, t, y), \Sigma_t).$$

where $p_\theta(z_{t-1}|z_t, y)$ represents the denoising distribution, parameterized by $\theta$, and is modeled as a Gaussian with mean $\mu_\theta(z_t, t, y)$ and covariance $\Sigma_t$. While $\mu_\theta$ is an output of the learned model, $\Sigma_t$ follows a predefined schedule, such that $\Sigma_0 < \Sigma_1 < \cdots < \Sigma_T$.

Given the complexity of directly computing the exact log-likelihood $\log(p_\theta(z_0|y))$ in the reverse process, it is common to use the ELBO (Kingma and Welling, 2013) as a tractable surrogate objective. Using properties of diffusion models, this ELBO formulation leads to a specific loss function that optimizes the noise-prediction model:

$$L_{LDM} = \mathbb{E}_{z, \epsilon \sim \mathcal{N}(0,1), t, y} \left[ ||\epsilon - \epsilon_\theta(z_t, t, y)||_2^2 \right],$$

where $t$ is uniformly distributed over $1, ..., T$, $\epsilon \sim \mathcal{N}(0, 1)$, and $\epsilon_\theta(z_t, t, y)$ is the predicted noise for computing $\mu_\theta(z_t, t, y)$. For details, see Ho et al. (2020).

## 2.2 U-NETWORKS

U-Nets, a CNN architecture, have demonstrated their effectiveness across a range of generative tasks, including image synthesis and restoration (Ronneberger et al., 2015; Isola et al., 2017).

A U-Net consists of a downsampling path (i.e. down-blocks), an upsampling path (i.e. up-blocks), and a mid-block. The downsampling path compresses the input $z_t$ into a latent representation $m_t^{\text{pre}} \in \mathbb{R}^{1280 \times 8 \times 8}$, where $\text{down}(z_t) = m_t^{\text{pre}}$, by reducing spatial dimensions and increasing the number of feature channels. The mid-block refines this latent representation into $m_t^{\text{post}} \in \mathbb{R}^{1280 \times 8 \times 8}$, where $\text{mid}(m_t^{\text{pre}}) = m_t^{\text{post}}$. The up-block then reconstructs the image by upsampling $m_t^{\text{post}}$ to $z_{t-1}$, the next latent representation in the denoising process. This process effectively combines low-level details with high-level semantic information.

U-Nets are widely used in diffusion-based generative models, where they model $\epsilon_\theta(z_t, t, y)$, effectively removing noise while preserving structure. The ability to maintain both local and global information through skip connections makes U-Nets particularly suited for diffusion models.

To then make our models conditional on a prompt $y$, we map $y$ through a tokenizer $\tau_\theta$ and pass this intermediate representation within the down-, mid- and up- blocks via a cross-attention layer $\text{Attention}(Q, K, V) = \text{softmax}\left(\frac{QK^T}{\sqrt{d}}\right) V$ (Vaswani et al., 2017). We mathematically denote this as follows:

$$Q = W_Q \phi_\theta(z_t), \quad K = W_K \tau_\theta(y), \quad V = W_V \tau_\theta(y).$$

Here, $W_Q, W_K$, and $W_V$ are learned projection matrices, and $\phi_\theta(z_t)$ and $\tau_\theta(y)$ represent the encoded latent representations of the inputs $z_t$ and $y$. The cross-attention output is then passed through a feed-forward neural network, as in the transformer architecture.

## 2.3 SPARSE MIXTURE OF EXPERTS

MoE is a widely-used machine learning architecture designed to handle complex tasks by combining the outputs of several specialized models, or "experts" (Jacobs et al., 1991). The key intuition behind MoE is that different experts can excel at solving specific parts of a problem, and by dynamically selecting or weighing their contributions, the MoE can perform more effectively.

Our pre-trained models contain sparse MoE layers which combine multiple expert models at cross-attention layers and feed-forward layers embedded within the U-Net architecture (Shazeer et al., 2017; Fedus et al., 2022). Let $M$ denote the number of experts, and let $i$ denote the $i$-th expert. The cross-attention layer can then be expressed as:

$$Q^i = W_Q^i \phi_\theta(z_t), \quad K^i = W_K^i \tau_\theta(y), \quad V^i = W_V^i \tau_\theta(y).$$

Figure 2: EMoE separates expert components in the first cross-attention layer in the first down-block and processes each component separately as an independent computation path in the MoE pipeline. This results in $M$ distinct latent representations after the first denoising step. The figure illustrates an ensemble with two expert components, (■ and ■).

The matrices $W_Q^i$, $W_K^i$, and $W_V^i$ are learned projection matrices specific to each expert $i$, allowing each expert to attend to different aspects of the input information.

A similar process occurs within the feed-forward networks, where each expert processes the data independently before their results are combined (Lepikhin et al., 2020). The ensemble created by this mechanism leads to more robust predictions, as each expert is able to specialize and contribute uniquely to the final output. In addition to the ensemble created by the cross-attention and feed-forward layers, the MoE architecture includes a routing or gating network that dynamically selects which experts to activate. The gating network determines the top $n \leq M$ experts to use for a given input, and the final output is computed as a weighted sum of the selected experts' outputs:

$$Q = \sum_{i \in \mathcal{S}} w^i Q^i, \quad K = \sum_{i \in \mathcal{S}} w^i K^i, \quad V = \sum_{i \in \mathcal{S}} w^i V^i, \tag{1}$$

where $\mathcal{S}$ is the set of selected experts, $w^i$ is the weight of the $i$-th expert. This combination of expert specialization and dynamic routing allows MoE models to scale efficiently by being sparse and only selecting a subset of experts to pass through.

## 3 EPISTEMIC MIXTURE OF EXPERTS

Epistemic uncertainty arises where models lack knowledge, often in regions underrepresented in training data (Gruber et al., 2023; Wang and Ji, 2024). EMoE extends mixture-of-experts diffusion models to capture this uncertainty by measuring variance across experts, following ensemble principles (Lakshminarayanan et al., 2017). Unlike standard MoEs that merge expert outputs, EMoE disentangles them, making disagreement directly observable. This disagreement serves as a robust signal of bias and coverage gaps in the training distribution.

### 3.1 SEPARATION OF EXPERTS

At the first sparse MoE layer, located in the initial cross-attention layer of the first down-block, we assign each expert to a distinct computational path instead of aggregating outputs via a weighted sum. Each path processes its own copy of the latent representations, and for the $i$-th expert, the output of the cross-attention mechanism is given by:

$$CA^i = \text{Attention}(Q^i, K^i, V^i).$$

These outputs propagate independently through the model along their designated paths. However, in later sparse MoE layers, expert outputs are aggregated within each computational path according to Equation 1. As a result, the model maintains $M$ distinct representations of the latent space $z_t$, rather than a single aggregated representation. Figure 2 illustrates the flow of this process within the model.

Separating the ensemble components early in the pipeline generates multiple predictions within the latent spaces of the denoising process. This enables the estimation of epistemic uncertainty at the initial step of denoising without requiring a complete forward pass through the U-Net, offering the

advantage of halting the denoising process immediately for uncertain prompts. Diffusion models carry the drawback of being computationally expensive during image generation. This limitation has spurred considerable research to accelerate the denoising process (Huang et al., 2022; Wu et al., 2023). The fast computation of epistemic uncertainty in our approach aligns with ongoing efforts to reduce the environmental impact of large machine learning models (Henderson et al., 2020).

## 3.2 EPISTEMIC UNCERTAINTY ESTIMATION

We capture epistemic uncertainty by measuring the variance among the ensemble components, a common approach in the literature (Ekmekci and Cetin, 2022; Chan et al., 2024). This occurs after the mid-block in our U-Net, $m_T^{post}$. Note that given that this is a high-dimensional space $d_{mid}$ with dimensions $1280 \times 8 \times 8$ and we want to reduce epistemic uncertainty to one number, we take the mean across the variance of each dimension. Thus our estimate of epistemic uncertainty is,

$$\text{EU}(y) = \mathbb{E}_{d_{mid}} \left[ \text{Var}_{i \in M} \left[ m_T^{post} \right] \right]. \quad (2)$$

It is important to note that $m_T^{post}$ takes as input the text prompt, $y$. Thus $\text{EU}(y)$ gives an estimate of the epistemic uncertainty of our MoE given a prompt $y$. The intuition behind this choice of epistemic uncertainty estimator is detailed in Appendix D. Note that the epistemic uncertainty values reflect the model's confidence in the generation, with higher uncertainty indicating that the model is extrapolating further from its training data. This provides valuable insight into the reliability of the generated content, particularly for rare or unseen prompts.

## 3.3 BUILDING MOE

To build an ensemble that effectively captures uncertainty, the ensemble components must be diverse enough to reflect meaningful disagreement among them. In deep learning, two primary techniques have been used to achieve diversity among ensemble components: bootstrapping samples during training and random initialization (Breiman, 2001; Lakshminarayanan et al., 2017). In our approach, we do not train the ensemble components; instead, they are sourced from pre-existing models available on Hugging Face and Civit AI. This strategy offers the significant advantage of enabling the creation of countless MoE, as Hugging Face hosts over 30,000 model checkpoints and Civit AI provides thousands of models.

Without controlling the training process, ensemble diversity relies largely on chance. Fortunately, platforms like Hugging Face and Civit AI offer a wide range of task-specific models, enhancing diversity. As shown in Section 4.4, even 4 checkpoints of the same models can suffice. In contrast, training such an ensemble from scratch would demand 150,000 GPU-hours per expert, costing $600,000 USD.

Finally, after assembling the ensemble, a gating module is needed to route inputs to a subset of experts and assign appropriate weights to their outputs. Although the gating module can be trained, it can also be inferred by computing the similarity of a given input with $\psi^i$, a text description characterizing the strengths and weaknesses of each expert. The gating weights are computed as follows:

$$\alpha^i = \tau_\theta(\psi^i) \cdot \tau_\theta(y), \qquad w^i = \text{softmax}(\alpha^i),$$

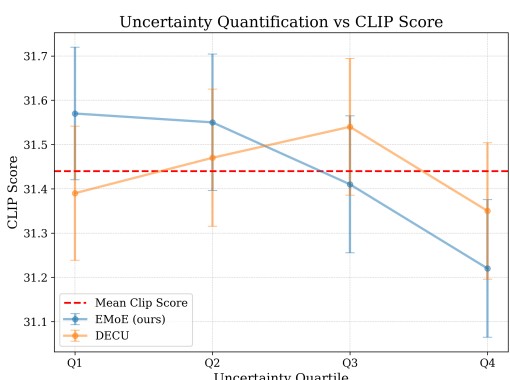

Figure 3: CLIP score across different uncertainty quartiles. EMoE accurately attributes prompts that produce images with high CLIP scores with low uncertainty unlike DECU. The red line indicates the average CLIP score across all quartiles.

Table 1: CLIP score, Aesthetic score & Image Reward on each uncertainty quartile, using EMoE, on the English 40k prompt dataset. Note the CLIP Scores are reported as $\mu \pm \sigma$.

| Quartile | CLIP Score ↑ | Aesthetic Score ↑ | Image Reward ↑ |
|---|---|---|---|
| Q1 | 31.578±0.15 | 5.763 | 0.292 |
| Q2 | 31.546±0.16 | 5.744 | 0.290 |
| Q3 | 31.405±0.16 | 5.733 | 0.273 |
| Q4 | 31.217±0.16 | 5.682 | 0.266 |

Table 2: Mean Length of English Prompts by Quartile of Uncertainty ± standard deviation.

| Quartile | Character Count | Word Count |
|---|---|---|
| Q1 | 53.14 ± 13.50 | 10.58 ± 2.56 |
| Q2 | 52.38 ± 12.94 | 10.47 ± 2.42 |
| Q3 | 52.20 ± 12.81 | 10.43 ± 2.39 |
| Q4 | 51.93 ± 12.32 | 10.34 ± 2.33 |

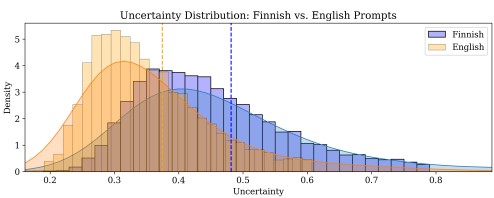

Table 3: Comparison of CLIP scores and mean uncertainty $\pm$ standard deviation between Finnish and English prompts.

| Language | CLIP Score $\uparrow$ | Uncertainty |
|----------|-----------------------|-------------|
| Finnish | 16.41 | $0.48 \pm 0.19$ |
| English | 31.39 | $0.37 \pm 0.14$ |

Figure 4: Uncertainty distribution for Finnish and English prompts, showing higher uncertainty for Finnish prompts compared to English.

where $\cdot$ represents the dot product. The gating module assigns weights to experts by computing the similarity between the input representation, $\tau_\theta(y)$, with the expert-specific representations, $\tau_\theta(\psi^i)$. This method enables the MoE model to dynamically select and weigh experts without requiring additional training. Further details can be found in Appendix E and in Goddard et al. (2024).

## 4 RESULTS

To validate EMoE, we conducted experiments on the COCO dataset (Lin et al., 2014), building on the diffusers and segmoe libraries (von Platen et al., 2022; Yatharth Gupta, 2024) with modifications to support our method. We used the base MoE provided in segmoe (Appendix F), which includes four experts. This choice avoids researcher bias from manual expert selection and ensures that the model, not originally designed for uncertainty estimation, provides a neutral testbed for EMoE. For multilingual evaluation, COCO prompts were translated using the Google Translate API.

We report results using CLIP score (Hessel et al., 2021), which measures semantic alignment between a generated image and its text prompt (higher is better). For non-English prompts, the English translation was used in scoring. We discuss the limitations of CLIP score as a metric in Appendix B.

### 4.1 ENGLISH PROMPTS

Our first experiment evaluated EMoE's ability to distinguish between in-distribution prompts that yield higher-quality images. We randomly sampled 40,000 COCO prompts and computed their epistemic uncertainty with EMoE. Prompts were then divided into quartiles: Q1 contained the lowest 25% of uncertainty values, through Q4, the highest 25%. For each quartile, we generated images and assessed quality using the CLIP score. As shown in Figure 3, we report the mean and standard deviation of CLIP scores across quartiles, revealing a clear trend: lower uncertainty corresponds to higher image quality. This demonstrates EMoE's effectiveness in fine-grained uncertainty estimation on in-distribution data—a capability that DECU (Berry et al., 2024) did not exhibit.

This relationship extends to other evaluation metrics. Table 1 shows a similar pattern across quartiles for Aesthetic Score and Image Reward (Schuhmann et al., 2022; Xu et al., 2024), further affirming EMoE's robustness in estimating uncertainty for MoE text-to-image models. The small differences between quartile scores reflect that all prompts in the English dataset are in-distribution.

We also analyzed prompt characteristics by uncertainty quartile. Prompts in lower uncertainty quartiles were longer in both character and word count, as shown in Table 2. This aligns with the intuition that longer, more descriptive prompts offer clearer objectives to the model.

To further validate these trends, we extended our analysis to the CC3M dataset (Sharma et al., 2018), as detailed in Appendix C. The results on CC3M further support our findings on the COCO dataset.

### 4.2 FINNISH PROMPTS

Next, we evaluated EMoE's ability to distinguish in-distribution from out-of-distribution samples by translating 10,000 English COCO prompts into Finnish. Since Finnish is less represented in online datasets, we expected these prompts to be more likely out-of-distribution and to yield lower-quality images. As shown in Figure 4, the uncertainty distribution for Finnish prompts is shifted to the right relative to English, confirming EMoE's capacity to detect distributional differences. To quantify

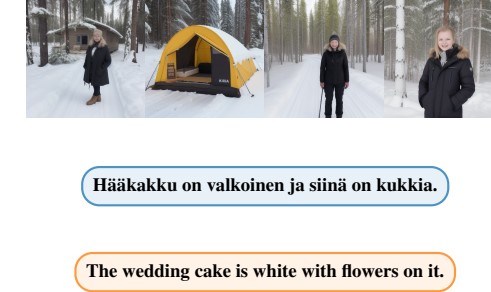

> Neliön muotoinen pizza, jonka henkilö leikkaa isolla veitsellä

> A square shaped pizza being cut by a person with a big knife.

> Hääkakku on valkoinen ja siinä on kukkia.

> The wedding cake is white with flowers on it.

Figure 5: Qualitative comparison of image-generation for a Finnish prompt with the word "pizza" and a random Finnish prompt. The English translation was not provided to the model.

Table 4: Mean Length of Finnish Prompts by Quartile of Uncertainty.

| Quartile | Character Count | Word Count |
|----------|-----------------|------------|
| Q1 | $54.94 \pm 17.04$ | $6.59 \pm 2.16$ |
| Q2 | $51.26 \pm 14.40$ | $6.14 \pm 1.79$ |
| Q3 | $49.67 \pm 14.23$ | $5.95 \pm 1.75$ |
| Q4 | $47.97 \pm 13.86$ | $5.77 \pm 1.73$ |

Table 5: Comparison of the proportion of prompts with "pizza" in Q1 of uncertainty between Finnish and English.

| Language | Proportion of Prompts with "pizza" in Q1 |
|----------|------------------------------------------|
| Finnish | 46.67% |
| English | 21.54% |

this effect, we report the area under the receiver operating characteristic curve (AUROC). Here, the AUROC represents the probability that a randomly chosen Finnish prompt (out-of-distribution) receives a higher uncertainty score than a randomly chosen English prompt (in-distribution). Higher values indicate better separation: perfect separation yields 1.0, while random guessing yields 0.5. EMoE achieves a strong AUROC of 0.745.

We also applied EMoE to probe model biases. In images generated from Finnish prompts, those containing the word "pizza" consistently produced more text-aligned outputs than random prompts (Figure 5). EMoE reflected this bias quantitatively: 46.67% of Finnish "pizza" prompts fell into the lowest-uncertainty quartile (Q1), compared to only 21.54% for English prompts (Table 5). Shared vocabulary between English and Finnish likely contributes to certain Finnish prompts exhibiting lower uncertainty than some English prompts.

Finally, we analyzed the relationship between prompt length and uncertainty. As shown in Table 3, longer prompts reduce uncertainty even in Finnish (Table 4), suggesting that added context promotes greater expert agreement across languages.

### 4.3 MULTI-LINGUAL PROMPTS

To further explore the behavior of EMoE, we translated 1,000 prompts into an additional 23 languages. We applied EMoE to these translations and calculated each language's respective CLIP score. As shown in Figure 6, there is a strong negative correlation ($r = -0.79$) between uncertainty and CLIP score, consistent with the expected relationship between uncertainty and image quality. Additionally, the size of each point in Figure 6 is proportional to the number of native speakers for each language. One can also observe a relationship between the number of native speakers with both CLIP score and uncertainty of any given language. European languages generally performed better than non-European languages, which further underscores the potential bias in favor of European languages in text-to-image models and EMoE's ability to capture language related model bias.

### 4.4 ABLATION

We conducted 4 ablation experiments to validate the robustness of our approach. All ablation studies were performed on the dataset of 40,000 English prompts.

To identify the optimal number of ensemble components, we examined ensemble sizes of 2 and 3, using all possible permutations from the 4 components. We averaged the results for ensembles of 2 and 3 components (Figure 7a). The results indicate that ensemble sizes of 2 and 3 are sub-optimal to an ensemble size of 4, as the first quantile (Q1) yields a lower CLIP score than the second (Q2).

We investigated the effect of the denoising step on uncertainty quantification, as shown in Figure 7b. A consistent decrease in CLIP scores across uncertainty quantiles at each step confirmed EMoE's robustness in estimating epistemic uncertainty. For practical reasons, we selected the first step, as it offers the earliest opportunity to halt the costly denoising process for high-uncertainty prompts.

We also explored different latent spaces in which to estimate epistemic uncertainty, testing both $Var(m_T^{pre})$ and $Var(z_{T-1})$. The results, shown in Figure 7c, indicate that $Var(z_{T-1})$ is sub-optimal, aligning with previous findings from DECU. We observed that $Var(m_T^{pre})$ performed similarly to $Var(m_T^{post})$. We chose $Var(m_T^{post})$ because the mid-block is intended to refine the latent space, though $Var(m_T^{pre})$ could serve as an acceptable alternative.

Finally, to further validate the robustness of EMoE, we ran an additional experiment using Runway MoE (Figure 7d). The results confirm that EMoE is versatile and can effectively handle different MoE models. Additionally, this demonstrates that EMoE can detect uncertainty even within for one model as each expert component is a different checkpoint of one model.

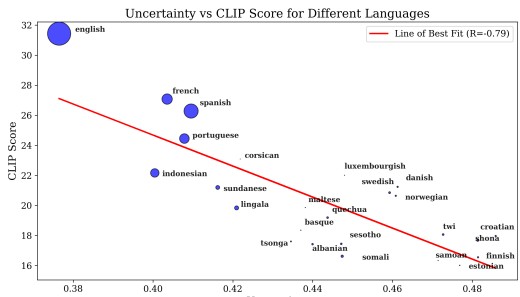

Figure 6: Negative correlation between uncertainty and image quality across prompts translated into 25 different languages. EMoE demonstrates a strong negative correlation (r = -0.79) between uncertainty and CLIP score, with languages having more native speakers generally producing lower uncertainty and higher-quality images, highlighting potential biases in text-to-image models favoring more commonly spoken languages.

## 5 RELATED WORKS

Building ensembles of diffusion models for image generation is challenging due to their large parameter sizes, often hundreds of millions (Saharia et al., 2022; Nichol et al., 2022; Ramesh et al., 2022). Despite this, models like eDiff-I enhance image fidelity using ensembles (Balaji et al., 2022), though they don't target epistemic uncertainty. DECU, by contrast, is designed specifically for such uncertainty estimation but requires 7 days of training (Berry et al., 2024). Our method reduces this burden to zero by leveraging pre-trained experts. EMoE further tackles the harder task of estimating epistemic uncertainty in text-to-image generation.

Previous work has explored epistemic uncertainty in neural networks, primarily in image classification, via Bayesian methods (Gal et al., 2017; Kendall and Gal, 2017; Kirsch et al., 2019), which deal with simpler, discrete outputs. Ensemble methods have also been used for uncertainty in regression (Lakshminarayanan et al., 2017; Choi et al., 2018; Chua et al., 2018; Depeweg et al., 2018; Postels et al., 2020; Berry and Meger, 2023a;b). Notably, Postels et al. (2020) and Berry and Meger (2023b) used Normalizing Flows (NFs) to model uncertainty efficiently, and Berry and Meger (2023a) extended this to a 257-dimensional space using Pairwise Difference Estimators. Our approach scales this to 700k-dimensional outputs in diffusion models with text prompts.

As interest in uncertainty estimation grows, many methods have emerged for image and text generation (Malinin and Gales, 2020; Berry et al., 2024; Chan et al., 2024; Liu et al., 2024). For instance, Chan et al. (2024) used hyper-networks for uncertainty in weather-predictive diffusion models, whereas EMoE derives uncertainty from pre-trained experts available online.

Some studies have used epistemic uncertainty to detect hallucinations in large language models (Verma et al., 2023). While EMoE could be adapted for hallucination detection in vision-language models, lack of transparency in training data makes this infeasible in our case.

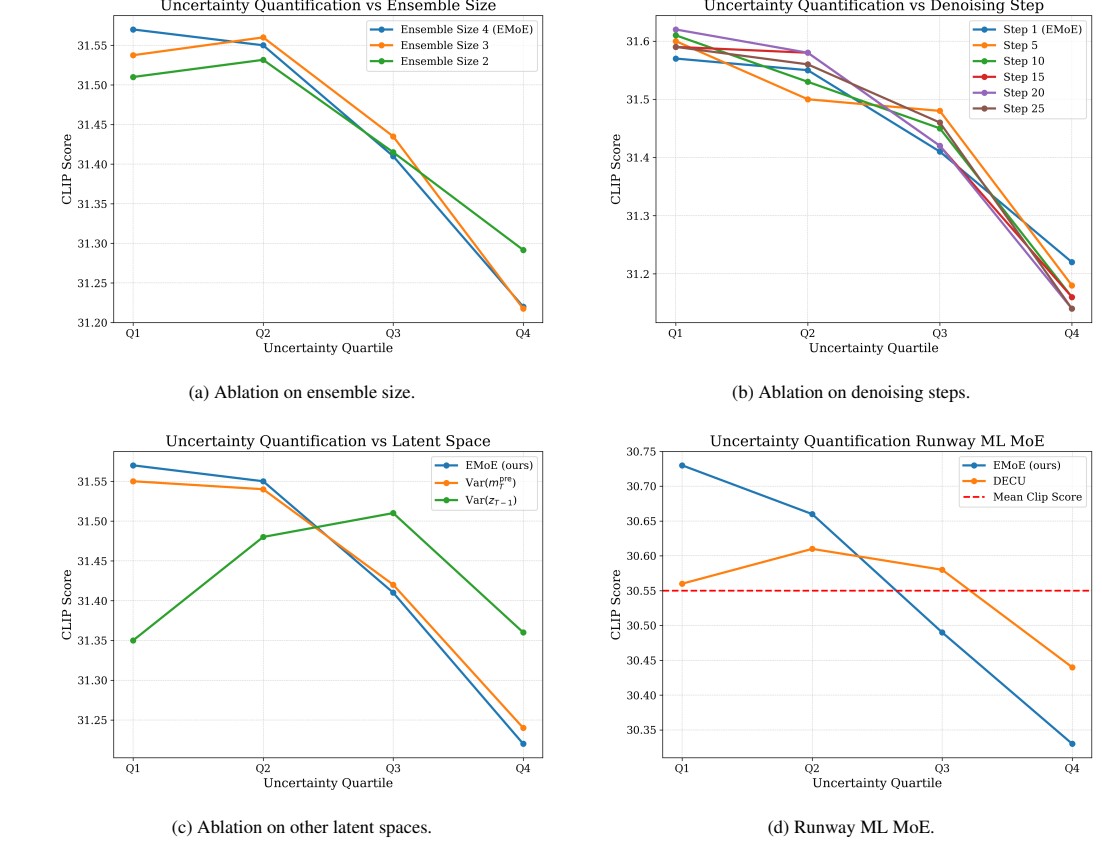

Figure 7: Ablation studies validating EMoE hyperparameters: ensemble size (a), denoising step (b), and latent space (c). Additionally, (d) shows the robustness of EMoE.

Past work has integrated uncertainty into MoE pipelines (Zheng et al., 2019; Luttner, 2023; Zhang et al., 2024), but without addressing epistemic uncertainty or text-to-image tasks. These methods are also not zero-shot, limiting their scope.

## 6 CONCLUSIONS

In this paper, we introduced EMoE which estimates uncertainty in text-to-image diffusion models. EMoE leverages pre-trained experts to provide computationally efficient uncertainty estimates without the need for training. By incorporating a latent space for uncertainty estimation within the diffusion process, EMoE can identify biases and uncertainty early in the generation process.

**Limitations**. While EMoE eliminates the need for additional training, it relies on the availability of pre-trained expert networks. Although publicly available models are abundant, they may not always offer sufficient diversity for optimal uncertainty estimation in all scenarios. Our results show that even with very similar experts (e.g., Runway ML MoE), EMoE produces reliable uncertainty estimates. However, this may not hold universally. Additionally, EMoE requires sufficient memory resources to load and execute an ensemble of experts efficiently.

Our experimental results demonstrate that EMoE enhances the detection of epistemic uncertainty while also exposing underrepresented linguistic biases in diffusion models. By leveraging readily available SOTA pre-trained models, we show that EMoE scales efficiently and delivers reliable uncertainty estimates across a range of input prompts. These capabilities have significant implications for fairness, accountability, and the robustness of AI-generated content.

As generative models continue to proliferate, the ability to quantify and interpret uncertainty will become increasingly critical, especially in applications involving autonomous decision-making. Future research may explore strategies to enhance expert diversity, optimize memory efficiency, and extend EMoE to more complex tasks and environments.

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

# A  COMPUTE DETAILS

We used the same set of hyperparameters as in the Stable Diffusion model described by Yatharth Gupta (2024). All experiments use DDIM sampling with 25 denoising steps and the standard $\epsilon$-prediction parameterization, where the model predicts the noise added to the latent state at each timestep, following the formulation in Rombach et al. (2022). Minor changes were made to both the Segmoe and Diffusers codebases to disentangle the MoE, with specific modifications to incorporate EMoE. Our infrastructure included an AMD Milan 7413 CPU running at 2.65 GHz, with a 128M L3 cache, and an NVIDIA A100 GPU with 40 GB of memory. The wall clock time required to collect each dataset and the memory usage are provided in Table 6. The parameter count for the Segmoe model is 1.63 billion parameters, while a single model contains 1.07 billion parameters. This highlights the efficiency of using a sparse MoE approach compared to creating 4 distinct models, as the Segmoe model is only 153% the size of a single model, rather than 400%. When running the SegMoE model in its standard mode, generating an image from one prompt takes an average of 3.58 seconds. In comparison, using EMoE typically requires an average of 12.32 seconds to generate four images from a single prompt. However, when only a single image per prompt is required, EMoE's output can be optimized by estimating epistemic uncertainty during the initial diffusion step. Once the uncertainty is determined, standard MoE-based image generation proceeds with the elimination of the unnecessary computational paths. This optimized version of EMoE, Fast EMoE, achieves an average generation time of 5.5 seconds. Table 7 provides further details. Note that uncertainty reported across all experiments is calculated as $\sqrt{d_{midsize}} \times \mathrm{EU}(y)$, where $d_{midsize} = 1280 \times 8 \times 8$.

Table 6: Computational requirements.

| Dataset | Run Time | Storage |
|---|---|---|
| English 40k Prompts | 200 gpu hrs | 6 TB |
| Finnish 10k Prompts | 50 gpu hrs | 1.5 TB |
| Other Languages 1k Prompts | 5 gpu hrs | 150 GB |

Table 7: Generation times for baseline (Segmoe) and two variants of EMoE. Reported times are $\mu \pm \sigma$.

| Model | Generation Time |
|---|---|
| Segmoe | $3.58 \pm 0.54$ secs |
| EMoE | $12.32 \pm 4.6$ secs |
| Fast EMoE | $5.5 \pm 0.15$ secs |

# B  BIAS IN CLIP SCORE

CLIP score, despite its known biases (Chinchure et al., 2023; Alabdulmohsin et al., 2024), remains a widely-used method for evaluating the alignment between text prompts and generated images, alongside FID (Shi et al., 2020; Kumari et al., 2023). Both metrics, however, rely on auxiliary models (CLIP and Inception, respectively), making them susceptible to inherent biases. FID requires a large number of samples for reliable estimation and thus requires more compute, whereas CLIP score facilitates a more direct assessment of text-to-image alignment with fewer samples (Kawar et al., 2023; Ho et al., 2022a). Moreover, FID does not measure text-image alignment, which is crucial in our context. Considering these trade-offs, we prioritized CLIP score due to its relevance to our research objectives and its broad acceptance in related studies.

To further validate our findings and address potential concerns regarding metric biases, we conducted additional experiments using the Aesthetic Score Predictor and Image Reward as evaluation metrics (Schuhmann et al., 2022; Xu et al., 2024). The Aesthetic Score Predictor quantifies how much people, on average, like an image, while Image Reward provides a score that encodes human preferences. The Aesthetic Score Predictor operates on a scale from 1 to 10, with higher scores indicating a more favored image, and higher Image Reward values similarly reflect images preferred by humans. Image Reward takes as input the prompt and the image, while Aesthetic Score only takes in the image as input.

Table 8: Comparison of Aesthetic Score and Image Reward with mean uncertainty $\pm$ standard deviation between Finnish and English prompts. Illustrating lower image quality and higher uncertainty for Finnish prompts.

| Language | Aesthetic Score ↑ | Image Reward ↑ | Uncertainty |
|---|---|---|---|
| Finnish | 5.917 | -2.143 | $0.48 \pm 0.19$ |
| English | 5.738 | 0.270 | $0.37 \pm 0.14$ |

---

**Algorithm 1** Epistemic Mixture of Experts (EMoE)

---

1: **Input:** Initial noise $z_T \sim \mathcal{N}(\mathbf{0}, \mathbf{I})$, total steps $T$, pre-trained experts $E = \{e_1, e_2, \ldots, e_M\}$, prompt $y$
2: **for** $t = T$ to 1 **do**
3:   **if** $t = T$ **then**
4:     **Separate Experts:**
5:     **for** each expert $e_i \in E$ **do**
6:       Pass $z_T$ and prompt $y$ through $e_i$'s first cross-attention layer $CA^i$ to arrive at $M$ distinct latent representations.
7:       Subsequent sparse MoE layers are processed as Equation 1.
8:       Extract the mid-block latent representation for each expert $m_T^{post,i}$.
9:     **end for**
10:    Compute epistemic uncertainty EU($y$) as defined in Equation 2.
11:    Output $M$ different $\mathbf{z}_{t-1}^i$, one for each expert.
12:   **else**
13:    **Mixture of Experts Rollout:**
14:    **for** $i \in \{1, ..., M\}$ **do**
15:     Update latent variable for each expert:

$$\mathbf{z}_{t-1}^i \sim p_\theta(\mathbf{z}_{t-1}^i | \mathbf{z}_t^i, y)$$

16:     Pass $\mathbf{z}_{t-1}^i$ and $y$ through our reverse diffusion process, as a standard MoE (e.g. experts are aggregated as Equation 1). This is shown in Figure 2 in ▇ and ▇.
17:    **end for**
18:   **end if**
19: **end for**
20: **Output:** $M$ reconstructed latent variables $\mathbf{z}_0^i$ and EU($y$).

---

We re-conducted a comparison between Finnish and English prompts to further evaluate EMoE's capabilities. Results from Image Reward align with the conclusion that EMoE effectively detects out-of-distribution data, as evidenced by lower Image Reward for Finnish prompts (Table 8). However, the Aesthetic Score does not support this conclusion. This discrepancy can be attributed to the nature of the Aesthetic Score, which evaluates the visual quality of the generated image independently of the text prompt. Consequently, it does not account for how well the image aligns with the prompt. The higher Aesthetic Score observed for the Finnish dataset can be explained by this limitation, as it overlooks the alignment challenges posed by out-of-distribution prompts.

Additionally, we evaluated the correlation between uncertainty and Image Reward across all languages, as shown in Figure 8. Consistent with expectations, a negative correlation is observed, where higher uncertainty is associated with lower image quality, as estimated by EMoE. This relationship also reveals a preference for European languages, with most points above the line of best fit corresponding to European languages. This observation further underscores potential biases in text-to-image models favoring European languages. EMoE's ability to capture these language-related biases, alongside its robust performance across diverse evaluation metrics, reinforces its capability to estimate epistemic uncertainty accurately and reliably.

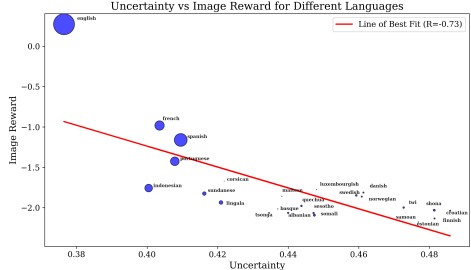

Figure 8: Negative correlation between uncertainty and image quality across prompts translated into 25 different languages. EMoE demonstrates a strong negative correlation (r = -0.73) between uncertainty and Image Reward, with languages having more native speakers generally producing lower uncertainty and higher-quality images, highlighting potential biases in text-to-image models favoring more commonly spoken languages.

## C   CC3M DATASET

To further validate our results, we extended the analysis of EMoE to the CC3M dataset (Sharma et al.,

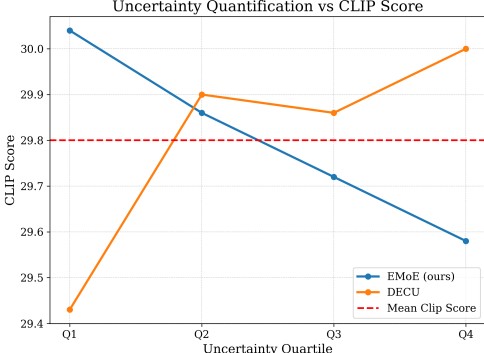

Figure 9: CLIP score on the CC3M dataset across different uncertainty quartiles. EMoE accurately attributes prompts that produce images with high CLIP scores with low uncertainty unlike DECU. The red line indicates the average CLIP score across all quartiles.

Table 9: CLIP score, Aesthetic score & Image Reward on each uncertainty quartile, using EMoE, on the English 10k prompt dataset for CC3M dataset.

| Quartile | CLIP Score ↑ | Aesthetic Score ↑ | Image Reward ↑ |
|---|---|---|---|
| Q1 | 30.07 | 5.75 | -0.05 |
| Q2 | 29.83 | 5.63 | -0.12 |
| Q3 | 29.72 | 5.60 | -0.17 |
| Q4 | 29.57 | 5.44 | -0.26 |

2018). The results are presented in Figure 9 and Figure 9. We randomly sampled 10,000 prompts from the dataset and repeated our analysis on English prompts. Our findings demonstrate that EMoE continues to perform robustly on the CC3M dataset. Specifically, prompts with lower uncertainty produced higher-quality images compared to those with higher uncertainty. These results not only reaffirm the robustness of our findings on the COCO dataset but also provide additional validation across a different dataset. Importantly, they suggest that EMoE's ability to capture biases is consistent and not merely an artifact of the COCO dataset. This further strengthens the argument for EMoE's effectiveness in diverse settings and its capacity for detecting biases in real-world data.

## D    INTUITION BEHIND OUR ESTIMATOR FOR EPISTEMIC UNCERTAINTY

Here is an intuitive explanation for our choice of estimator for epistemic uncertainty using the theory of Gaussian Processes. Each expert can be viewed as a sample from the posterior distribution of functions given an input $y$, denoted as $p(f(y)|y)$. By calculating the variance across these experts, we obtain the variance $\sigma^2$ of $p(f(y)|y)$, which serves as an estimate of epistemic uncertainty within the Gaussian Process framework. In general, other works have used the difference among ensemble components to denote epistemic uncertainty (Gal et al., 2017; Depeweg et al., 2018; Berry and Meger, 2023b).

When estimating the epistemic uncertainty for a prompt $y$, we weight each ensemble component equally. Therefore, let $\mathcal{F} = \{f_{\theta_i}\}_{i=1}^N$ denote an ensemble of $N$ neural networks, where each model $f_{\theta_i} : \mathcal{Y} \to \mathbb{R}$ is parameterized by $\theta_i$, sampled from a parameter distribution $p(\theta)$. Then the prediction from our ensemble is:

$$\hat{f}(y) = \frac{1}{N} \sum_{i=1}^{N} f_{\theta_i}(y),$$

where $y \in \mathcal{Y}$ is an input from the input space $\mathcal{Y}$.

A **Gaussian Process** (GP) is defined as a collection of random variables, any finite subset of which follows a joint Gaussian distribution. Formally, a Gaussian Process $f(y) \sim \mathcal{GP}(\mu(y), k(y, y'))$ is characterized by its mean function $\mu(y)$ and covariance function $k(y, y')$:

$$\mu(y) = \mathbb{E}[f(y)], \quad k(y, y') = \mathbb{E}[(f(y) - \mu(y))(f(y') - \mu(y'))].$$

**Proposition 1:** Let $\mathcal{F} = \{f_{\theta_i}\}_{i=1}^N$ be an ensemble of neural networks with parameter samples $\theta_i \sim p(\theta)$. As $N \to \infty$ and under the assumption that the neural network weights are drawn i.i.d. from a distribution with zero mean and finite variance, the ensemble predictor $\hat{f}(y)$ converges in

distribution to a Gaussian Process:

$$\hat{f}(y) \xrightarrow{d} \mathcal{GP}(\mu(y), k(y, y')),$$

where $\mu(y)$ is the expected value of the ensemble output, and $k(y, y')$ is the covariance function defined by the variance of the ensemble.

**Proof:**

To prove this, we proceed in two main steps:

STEP 1: CONVERGENCE OF MEAN FUNCTION

Consider the mean function $\mu(y)$ of the ensemble predictor:

$$\mu(y) = \mathbb{E}_{\theta \sim p(\theta)}[f_\theta(y)].$$

As $N \to \infty$, by the law of large numbers, the empirical mean of the ensemble $\hat{f}(y)$ converges to the expected mean:

$$\lim_{N \to \infty} \frac{1}{N} \sum_{i=1}^{N} f_{\theta_i}(y) = \mu(y).$$

STEP 2: CONVERGENCE OF COVARIANCE FUNCTION

The covariance function $k(y, y')$ of the Gaussian Process can be defined as:

$$k(y, y') = \lim_{N \to \infty} \frac{1}{N} \sum_{i=1}^{N} (f_{\theta_i}(y) - \mu(y)) (f_{\theta_i}(y') - \mu(y')).$$

Under the assumption that $f_{\theta_i}(y)$ are i.i.d. samples with finite variance, by the Central Limit Theorem (CLT), the ensemble prediction $\hat{f}(y)$ converges in distribution to a Gaussian Process $\mathcal{GP}(\mu(y), k(y, y'))$.

In the context of an ensemble of neural networks, **epistemic uncertainty** arises from the uncertainty over the model parameters $\theta$. This uncertainty is captured by the variance of the ensemble predictions:

$$\mathrm{Var}[\hat{f}(y)] = \frac{1}{N} \sum_{i=1}^{N} (f_{\theta_i}(y) - \hat{f}(y))^2.$$

As $N \to \infty$, this variance converges to the posterior variance of the Gaussian Process:

$$\lim_{N \to \infty} \mathrm{Var}[\hat{f}(y)] = k(y, y),$$

where $k(y, y)$ is the marginal variance of the Gaussian Process and directly represents the **epistemic uncertainty**.

## E  GATES WITHOUT TRAINING

Each expert is associated with a positive and a negative descriptor, $\psi^i = \left(p^i, n^i\right)$, which represent what the expert excels at and struggles with modeling, respectively. These descriptors are processed through a pre-trained text model, $\tau_\theta$, to create *gate vectors*, $v^i = [\tau_\theta(p^i); \tau_\theta(n^i)]$. When a new positive and negative prompt, $y_j = (\phi_j, \nu_j)$, is provided to generate an image, the latent representation of these prompts, $l_j[\tau_\theta(\phi_j); \tau_\theta(\nu_j)]$ are compared against $v^i$ and assigned a weight, $w_i^j$ based on the dot product and a softmax. This process is illustrated in Figure 10 and described in Goddard et al. (2024).

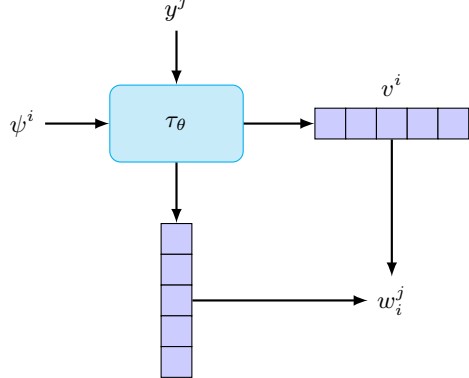

Figure 10: This pictures depicts how to have accurate gates without training.

## F  MODEL CARDS

Below are the model parameters for the base Segmoe MoE used in the experiments. We increased the number of experts from 2 to 4 to incorporate more ensemble components. Generally, having a low number of ensemble components (2-10) is sufficient in deep learning to capture model disagreement (Osband et al., 2016; Chua et al., 2018; Fujimoto et al., 2018). In addition to the Segmoe base MoE, we also tested EMoE on another MoE model, referred to as Runway ML, where each expert component is a Runway model. The corresponding model card can be found below. This experiment demonstrates the robustness of EMoE across different MoE architectures, showing that EMoE is effective even when components are trained on similar data with similarly initialized weights, as each Runway ML component was fine-tuned on new data from similar initial conditions.

```
Segmoe MoE

base_model: SG161222/Realistic_Vision_V6.0_B1_noVAE
num_experts: 4
moe_layers: all
num_experts_per_tok: 2
type: sd
experts:
  - source_model: SG161222/Realistic_Vision_V6.0_B1_noVAE
    positive_prompt: "cinematic, portrait, photograph, instagram,
        fashion, movie, macro shot, 8K, RAW, hyperrealistic, ultra
         realistic,"
    negative_prompt: " (deformed iris, deformed pupils, semi-
        realistic, cgi, 3d, render, sketch, cartoon, drawing,
        anime), text, cropped, out of frame, worst quality, low
        quality, jpeg artifacts, ugly, duplicate, morbid,
        mutilated, extra fingers, mutated hands, poorly drawn
        hands, poorly drawn face, mutation, deformed, blurry,
        dehydrated, bad anatomy, bad proportions, extra limbs,
        cloned face, disfigured, gross proportions, malformed
        limbs, missing arms, missing legs, extra arms, extra legs,
         fused fingers, too many fingers, long neck"
  - source_model: dreamlike-art/dreamlike-anime-1.0
    positive_prompt: "photo anime, masterpiece, high quality,
        absurdres, 1girl, 1boy, waifu, chibi"
    negative_prompt: "simple background, duplicate, retro style,
        low quality, lowest quality, 1980s, 1990s, 2000s, 2005
        2006 2007 2008 2009 2010 2011 2012 2013, bad anatomy, bad
        proportions, extra digits, lowres, username, artist name,
        error, duplicate, watermark, signature, text, extra digit,
         fewer digits, worst quality, jpeg artifacts, blurry"
  - source_model: Lykon/dreamshaper-8
    positive_prompt: "bokeh, intricate, elegant, sharp focus, soft
         lighting, vibrant colors, dreamlike, fantasy, artstation,
         concept art"
    negative_prompt: "low quality, lowres, jpeg artifacts,
        signature, bad anatomy, extra legs, extra arms, extra
        fingers, poorly drawn hands, poorly drawn feet, disfigured
        , out of frame, tiling, bad art, deformed, mutated, blurry
        , fuzzy, misshaped, mutant, gross, disgusting, ugly,
        watermark, watermarks"
  - source_model: dreamlike-art/dreamlike-diffusion-1.0
    positive_prompt: "dreamlikeart, a grungy woman with rainbow
        hair, travelling between dimensions, dynamic pose, happy,
        soft eyes and narrow chin, extreme bokeh, dainty figure,
        long hair straight down, torn kawaii shirt and baggy jeans
        , In style of by Jordan Grimmer and greg rutkowski, crisp
        lines and color, complex background, particles, lines,
        wind, concept art, sharp focus, vivid colors"
    negative_prompt: "nude, naked, low quality, lowres, jpeg
        artifacts, signature, bad anatomy, extra legs, extra arms,
         extra fingers, poorly drawn hands, poorly drawn feet,
        disfigured, out of frame"
```

```
Runway ML MoE

base_model: runwayml/stable-diffusion-v1-5
num_experts: 4
moe_layers: all
num_experts_per_tok: 4
type: sd
experts:
  - source_model: runwayml/stable-diffusion-v1-5
    positive_prompt: "ultra realistic, photos, cartoon characters,
        high quality, anime"
    negative_prompt: "faces, limbs, facial features, in frame,
        worst quality, hands, drawings, proportions"
  - source_model: CompVis/stable-diffusion-v1-4
    positive_prompt: "ultra realistic, photos, cartoon characters,
        high quality, anime"
    negative_prompt: "faces, limbs, facial features, in frame,
        worst quality, hands, drawings, proportions"
  - source_model: CompVis/stable-diffusion-v1-3
    positive_prompt: "ultra realistic, photos, cartoon characters,
        high quality, anime"
    negative_prompt: "faces, limbs, facial features, in frame,
        worst quality, hands, drawings, proportions"
  - source_model: CompVis/stable-diffusion-v1-2
    positive_prompt: "ultra realistic, photos, cartoon characters,
        high quality, anime"
    negative_prompt: "faces, limbs, facial features, in frame,
        worst quality, hands, drawings, proportions"
```

## G  STATISTICAL ANALYSIS

For Table 1 and Table 2, we used the Jonckheere-Terpstra test, which is the most appropriate choice for our data. This non-parametric test specifically tests for an ordered trend across multiple groups (i.e., whether the means follow a consistent ordering, such as $\mu_1 \geq \mu_2 \geq \mu_3 \geq \mu_4$). Given that we are testing for trends rather than just differences between groups, the Jonckheere-Terpstra test is suitable.

For the CLIP Score Table 1, the test yielded a p-value of $3.34 \times 10^{-19}$. For Table 2, the test yielded a p-value of $2.13 \times 10^{-12}$ for prompt length and $1.23 \times 10^{-7}$ for word count. Additionally, we used a t-test to compare the means for Finnish and English (i.e., $\mu_{\text{Finnish}} > \mu_{\text{English}}$) and obtained a p-value of $9.51 \times 10^{-66}$. All of these results further confirm the statistical significance of our findings.

## H  QUALITATIVE RESULTS

In addition to the examples provided in the main paper, we have included additional qualitative results of our MoE model. Figure 11 shows two sets of images: low uncertainty images on the left and high uncertainty images on the right. Each row corresponds to a single prompt, while the columns display the outputs from different ensemble components. The low uncertainty prompts exhibit less variation across ensemble outputs, whereas the high uncertainty prompts show greater diversity, indicating the model's difficulty in capturing the semantic meaning of the prompt in the generated images. Here, we present another example of models showing bias towards Finnish prompts containing "pizza", as illustrated in Figure 12.

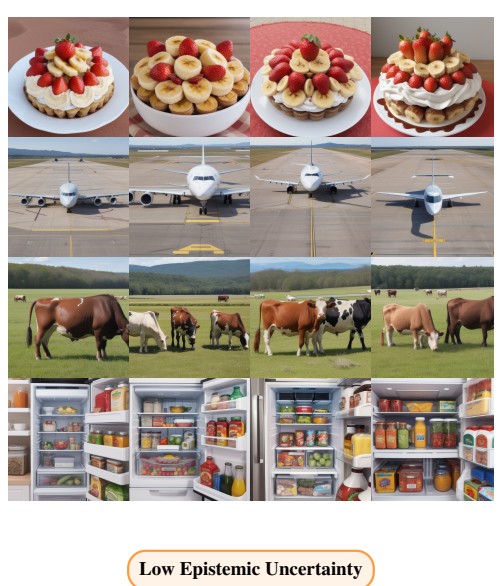 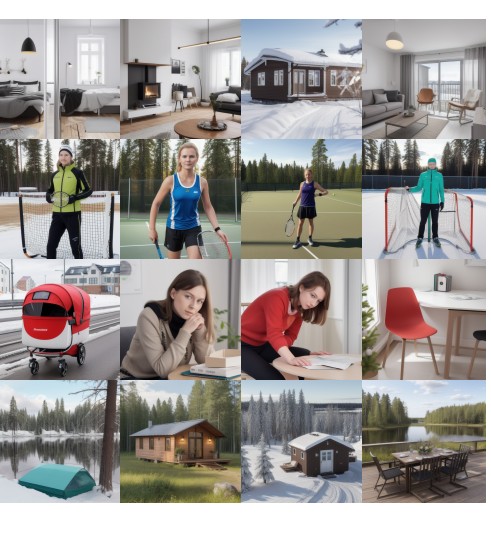

Low Epistemic Uncertainty  High Epistemic Uncertainty

Figure 11: EMoE's uncertainty across different prompts: Each row represents a distinct prompt, while the columns denote the output of each component. The left panel displays low uncertainty, while the right panel shows higher uncertainty, indicating more ambiguous or less familiar prompts.

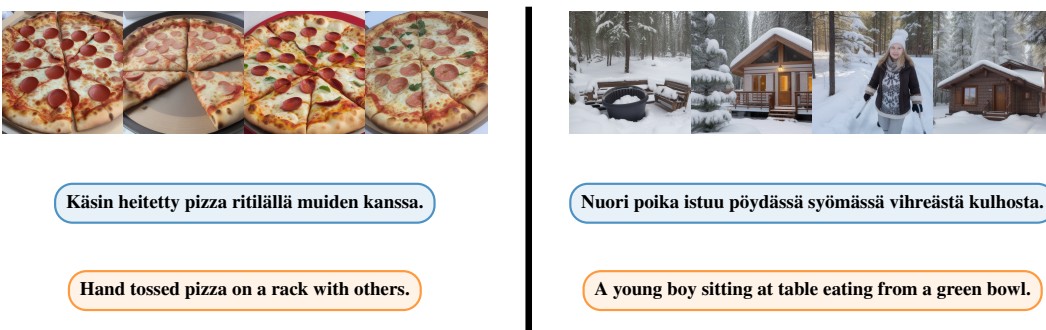

Käsin heitetty pizza ritilällä muiden kanssa.  Nuori poika istuu pöydässä syömässä vihreästä kulhosta.

Hand tossed pizza on a rack with others.  A young boy sitting at table eating from a green bowl.

Figure 12: Qualitative comparison of image-generation for a Finnish prompt with the word "pizza" and a random Finnish prompt. Note that the English translation was not provided to the model.

