# OpenReview forum: "Seeing the Unseen: How EMoE Unveils Bias in Text-to-Image Diffusion Models"
_ICLR.cc/2026/Conference — ICLR 2026 Conference Withdrawn Submission_

### Official Review · Reviewer_f7ee · 2025-10-29

**Soundness:** 2
**Presentation:** 2
**Contribution:** 2
**Rating:** 2
**Confidence:** 3

**Summary:**

This paper proposes EMoE (Epistemic Mixture of Experts), a framework for estimating epistemic uncertainty in text-to-image diffusion models without additional training. It leverages the diversity of pre-trained Mixture-of-Experts  architectures to provide zero-shot uncertainty estimation directly from prompts, operating efficiently in the latent space of the diffusion process. EMoE can identify underrepresented languages and regions in the training data, revealing hidden dataset biases and improving model transparency. Experiments on the COCO dataset show that EMoE yields reliable uncertainty estimates and outperforms existing approaches in detecting data imbalance.

**Strengths:**

1.	Their EMoE design does not require further training and estimates uncertainty in the latent space of the denoiser, enabling early detection of undersampled prompts before image generation is complete which makes it more applicable to real-world scenarios.
2.	They conduct thorough experiments over different language prompts to prove their framework.

**Weaknesses:**

1.	A potential weakness of this method is that the variance across experts may not purely reflect epistemic uncertainty. In DMs, cross-attention modules inherently promote generation diversity, which can inflate variance even for well-understood inputs. As a result, the estimated uncertainty may conflate true model epistemic uncertainty with normal semantic diversity.
2.	The output differences between different expert modules may do not necessarily indicate model uncertainty, rather, to some extent, they can reflect diversity and robustness in the generation process? For this point, a more explainable discussion is needed.
3.	Although their uncertainty estimation has clear indication trend that lower uncertainty corresponds to higher image quality, their estimation metric computes per-dimension variance followed by mean aggregation, which assumes independence across feature dimensions but diffusion latents are highly correlated and semantically heterogeneous, so the resulting value may not accurately reflect epistemic uncertainty.
4.	The variance of latent features is influenced by the prompt itself: concrete and simple prompts produce consistent patterns, while abstract prompts naturally yield higher variance. Thus, this metric conflates prompt diversity with model uncertainty, making it unclear whether high values indicate true epistemic uncertainty or inherently ambiguous prompts.
5.	The proposed uncertainty metric measures variance across different diffusion models rather than within a single model. Consequently, it reflects cross-model disagreement for a given prompt, which is heavily influenced by differences in model effectiveness, rather than the true epistemic uncertainty of a single target diffusion model. This may lead to overestimating uncertainty for prompts where an individual model is confident.

**Questions:**

please check the weakness

---

### Official Review · Reviewer_re8T · 2025-10-31

**Soundness:** 3
**Presentation:** 3
**Contribution:** 3
**Rating:** 6
**Confidence:** 3

**Summary:**

This paper introduces Epistemic Mixture of Experts (EMoE), a framework for estimating epistemic uncertainty in text-to-image diffusion models without additional training. EMoE separates pre-trained MoE expert pathways, computes variance across their latent representations in the diffusion denoiser, and interprets this as epistemic uncertainty. It identifies underrepresented prompts (e.g., non-English languages) where uncertainty is higher, revealing latent biases in training data. Experiments on COCO and CC3M datasets, covering English, Finnish, and 25 other languages, show the superiority of EMoE.

**Strengths:**

1. EMoE provides a novel approach to estimating epistemic uncertainty in diffusion models without additional training.

2. EMoE addresses transparency and bias detection in black-box generative models, a timely and socially relevant problem.

3. EMoE avoids training overhead (unlike DECU), showing computational savings and environmental benefits.

2. EMoE demonstrates superior performance on the COCO dataset compared to existing methods.

3. EMoE provides ablation studies that analyze design choices and confirm the robustness of its components.

**Weaknesses:**

1.The paper lacks a detailed comparison with other uncertainty estimation techniques beyond the COCO dataset, since this paper is mainly on DECU.

2.The scalability of EMoE to larger and more complex diffusion models is not thoroughly explored.

3.This paper only focus on COCO and CC3M (in appendix), which should be verified as evidence by evaluating other datasets.

4. Heavy reliance on CLIP-based metrics (known to favor English-centric datasets) may reinforce the very bias the paper aims to measure.

5. While EMoE correlates uncertainty with language underrepresentation, it stops short of quantitatively validating why or where this bias arises.

6. Some papers might be relavent to the topic, for data imbalance of diffusion trainnig.

[1] Class-Balancing Diffusion Models. CVPR 2023.

[2] PoGDiff: Product-of-Gaussians Diffusion Models for Imbalanced Text-to-Image Generation. NeurIPS 2025.

**Questions:**

1. Can you analyze the failure cases of EMoE, if possible?

2. How does the latent-space variance (Eq. 2) correlate numerically with epistemic uncertainty obtained from a gold-standard Bayesian ensemble or Monte Carlo dropout?

3. How sensitive is EMoE to the choice of checkpoint diversity? What happens if all experts are near-identical fine-tunes?

4. How does the performance of EMoE compare to other uncertainty estimation techniques across a wider range of datasets and tasks?

5. Can EMoE estimate aleatoric uncertainty as well, or only epistemic?

6. Have you verified whether the CLIP bias affects EMoE’s cross-lingual conclusions? (e.g., using multilingual CLIP or FID-based checks)

---

### Official Review · Reviewer_N3ww · 2025-11-01

**Soundness:** 3
**Presentation:** 3
**Contribution:** 2
**Rating:** 2
**Confidence:** 3

**Summary:**

This paper proposes EMoE, a zero shot method that uses MoE diffusion models to estimate epistemic uncertainty by measuring the variance across disentangled expert components in the latent space. This allows to detect underrepresented inputs in the train data. The key contributions are:
- The EMoE method, which allows zero shot uncertainty estimation by separating expert computational paths and measuring the variance in the latent space.
- EMoE uncertainty is shown to be consistent with expectations on in-distribution data.
- Detect novel and OOD data by quantifying uncertainty and comprehensive ablation studies.

**Strengths:**

- Zero shot uncertainty estimation.
- Consistent on in-distribution data shows reliability.
- Effective bias and OOD detection that can be used to enhance training.
- Detailed ablation studies and qualitative examples.

**Weaknesses:**

- The experimental section's findings are largely correlational and serve only to re-confirm established phenomena in generative modeling (e.g., uncertainty's inverse correlation with image quality, OOD inputs increase uncertainty, descriptive prompts lower uncertainty). The paper successfully proves EMoE works, but fails to provide new scientific insight.
- "halting the denoising process immediately for uncertain prompts". This is a strong case for the impact of the paper yet the paper contains no quantitative analysis of this tradeoff.
- While the paper mentions DECU there is a lack of comparison with existing literature and naive methods (e.g. simple latent-space reconstruction error, variance across diffusion steps).
- Observations like the Finnish prompts containing the word "pizza" fall into the lowest uncertainty quartile, are addressed with only a superficial explanation (shared vocabulary). This misses the opportunity to for a deeper analysis or visualization of the latent space.

**Questions:**

- Please provide a quantitative analysis of the proposed "early halting" feature. This can become a key contribution to the paper. How does the set uncertainty threshold relate to performance and computational resources?
- The findings on language bias are strong, but do they extend to other biases? These are very basic and well known relationships but this approach gives the unique opportunity to compare across weights and examine a lot more nuanced differences.
- To show the method works specifically for the MoE architecture and its not simply an effect of different models the authors must demonstrate that the uncertainty provided is more reliable or better than independently trained models.

---

### Note · Authors · 2025-11-17

**Comment:**

We thank the reviewers for their time and effort in evaluating our submission. After careful consideration, we have decided to withdraw our manuscript at this time.

**Withdrawal Confirmation:**

I have read and agree with the venue's withdrawal policy on behalf of myself and my co-authors.